# Therapeutic Exercise Prescription for Overhead Athletes with Shoulder Impingement Syndrome: A Systematic Review and CERT Analysis

**DOI:** 10.3390/jcm14051657

**Published:** 2025-02-28

**Authors:** Fabien Guérineau, María Dolores Sosa-Reina, Jaime Almazán-Polo, Javier Bailón-Cerezo, Ángel González-de-la-Flor

**Affiliations:** 1Department of Physiotherapy, Faculty of Medicine, Health and Sport, European University of Madrid, Villaviciosa de Odón, C/Tajo s/n, 28670 Madrid, Spain; fabien.guerineau@universidadeuropea.es (F.G.); mariadolores.sosa@universidadeuropea.es (M.D.S.-R.); angel.gonzalez@universidadeuropea.es (Á.G.-d.-l.-F.); 2CranioSPain Research Group, Centro Superior de Estudios Universitarios La Salle, 28023 Madrid, Spain; bailonfisioterapia@gmail.com

**Keywords:** overhead athletes, shoulder impingement syndrome, therapeutic exercise

## Abstract

**Background:** Shoulder impingement syndrome (SIS) is a prevalent condition among overhead athletes, often managed through therapeutic exercise interventions. However, the quality of reporting in exercise protocols significantly impacts their reproducibility and clinical implementation. The Consensus for Exercise Reporting Template (CERT) provides a standardized framework to assess the quality of exercise reporting in clinical research. **Objectives:** This systematic review aimed to evaluate the quality of exercise protocols used to treat SIS in overhead athletes by applying the CERT checklist. Additionally, the risk of bias was assessed to determine the methodological rigor of included studies. **Methods:** A systematic review was conducted following PRISMA guidelines. Six electronic databases (MEDLINE, CINAHL, Sport Discuss, Web of Science, and Cochrane) were searched for eligible studies. Inclusion criteria encompassed randomized controlled trials (RCTs), cohort studies, and case series that investigated exercise therapy for SIS in overhead athletes. Studies had to be published in English and provide details on exercise interventions. Exclusion criteria included non-human studies, acute injuries, and postoperative management. The primary outcome was the quality of intervention reporting, assessed using the CERT checklist. The secondary outcome was the risk of bias, evaluated using the modified Downs and Black checklist. **Results:** Five studies met the inclusion criteria, comprising four RCTs and one case series. CERT scores ranged from 6 to 13 (median = 8, IQR = 1), indicating suboptimal reporting quality. Commonly reported CERT items included equipment usage and exercise tailoring. However, key aspects such as adherence, motivation, and intervention fidelity were consistently underreported. None of the included studies provided comprehensive details on exercise interventions as per CERT guidelines, limiting their reproducibility and clinical application. **Conclusions:** The quality of reporting on exercise-based interventions for SIS in overhead athletes remains insufficient. Critical gaps in adherence monitoring, patient motivation, and intervention fidelity were identified. Future research should prioritize standardized and detailed reporting of exercise interventions to enhance reproducibility and facilitate evidence-based clinical practice.

## 1. Introduction

Shoulder impingement syndrome (SIS) is a concept that encompasses several disorders of the shoulder joint complex. Two types of SIS have been described: primary impingement, which corresponds to a structural modification of the subacromial space, resulting in a reduction in the latter; secondary impingement, on the other hand, corresponds to the same mechanism without structural modification [1]. The SIS is the result of several disorders, such as rotator cuff pathology, shoulder instability, scapular dyskinesis, biceps pathology, and glenohumeral internal rotation deficit [1]. SIS presents with painful symptoms in horizontal abduction with maximal external rotation and a certain amount of abduction of the shoulder, depending on the specific sports discipline [1]. Among overhead athletes, the incidence rate of SIS ranges from 0.2 to 1.8 cases per 1000 h of practice, while the prevalence of injury from SIS ranges from 5 to 36% [2].

Therapeutic exercise is the recommended treatment for people with a diagnosis of SIS and is effective in reducing pain and disability with a wide range of effect sizes [3]. Additionally, it is a cost-effective intervention [4]. Furthermore, it is an essential component of the rehabilitation process for overhead athletes, whether undergoing management alone or in conjunction with surgical intervention [5]. Nevertheless, the optimal exercise type, dosage, duration, and expected outcomes remain to be established [6].

In order to establish these recommendations, it is essential that interventions based on therapeutic exercises are described in sufficient detail. The quality of a study depends on the detail with which its intervention is described [7]. Moreover, a lack of detail makes it impossible to reproduce and apply these protocols [8]. In order to enhance the quality of the description of exercise-based interventions, the Consensus for Exercise Reporting Template (CERT) was developed in 2016 [9]. The CERT checklist can also be employed to evaluate the quality of published interventions. This checklist evaluates essential components of exercise interventions, including exercise intensity, frequency, and individualization, which are critical for ensuring effective rehabilitation. However, it remains unclear whether these key aspects are adequately reported in studies addressing SIS in overhead athletes.

In 2018, a systematic review by Wright et al. aimed to synthesize the use of therapeutic exercises for the treatment of shoulder pathology in overhead athletes [10]. Their conclusion was that the prescription of therapeutic exercises for shoulder injuries is based on expert opinion and that only one level B study recommends the use of single-plane, open-chain upper extremity exercises performed below 90° of elevation and closed-chain upper extremity exercises [10]. This underscores the need for a standardized, evidence-based approach to exercise prescription, which our study aims to address through the systematic evaluation of exercise interventions using the CERT checklist. Furthermore, the Bern 2022 consensus proposes a series of criteria, illustrated by exercises, to guide professionals working with overhead athletes in the management of shoulder injuries in overhead athletes. However, it concludes that these recommendations require updating by studies to assist practitioners in their decision-making [11].

To the best of our knowledge, no systematic review has assessed the quality of interventions using exercise therapy for the treatment of SIS in overhead athletes. It is, therefore, important to ascertain the level of quality of exercise interventions to improve and apply them in clinical practice. The aim of this study is, therefore, to assess the quality of existing exercise protocols through the CERT checklist for the treatment of SIS in overhead athletes.

## 2. Methods

This systematic review, conducted in accordance with the Preferred Reporting Items for Systematic Reviews and Meta-Analyses (PRISMA) guidelines (Appendix A), was preregistered on 28 July 2024 in the PROSPERO database under the identifier CRD42024570197.

### 2.1. Literature Search

A systematic search in the MEDLINE, CINAHL, Sport Discuss, Web of Science, and Cochrane databases was conducted by a physical therapist with experience in systematic reviews (FG) through 6 June 2024. The search strategy was tailored to each database in combination with sport-specific search terms combined with population and intervention terms (Appendix A).

### 2.2. Data Extraction

The flowchart illustrates the process followed in selecting the studies (Figure 1). Initially, duplicate studies captured from different databases were removed. To identify relevant articles, the titles and abstracts of all citations captured from the databases were independently screened by at least two authors (FG, AGF) based on the specified inclusion and exclusion criteria. Full-text articles were retrieved if the abstract did not provide enough information to determine eligibility or if the article passed the initial eligibility screening. If there was disagreement about whether to include an article, the three authors discussed the issue to reach a consensus. In this phase, the full texts of the studies that passed the previous stages were read, and those that met all the inclusion criteria of this review were selected. The selection criteria were then independently applied by two authors (FG, AGF) to the full text of the articles that passed the initial screening. Disagreements were resolved through discussion, and if consensus could not be reached, a third author (JAP) made the final decision.

### 2.3. Study Selection

To be included in the review, studies had to be randomized controlled trials (RCTs), cohort studies, or case series studies that involved overhead athletes with secondary impingement-related shoulder pain that had persisted for a minimum of one month. Additionally, the athletes in the studies had to have presented a positive test result in at least two of the following five tests proposed by Cools et al. [1]: Neer, Hawkins, Jobe, apprehension, and relocation tests. Furthermore, the athletes had to be between the ages of 17 and 45 years. The studies were required to include a description of the exercise therapy details, such as the treatment modality, prescription, type, and/or duration. The studies must be available in full-text and published in the English language.

Cadaver or non-human studies were not considered to be included. Studies with individuals with confirmed shoulder osteoarthritis; total shoulder arthroplasty; acute shoulder injuries, such as humerus, clavicle, or scapula fractures; and extra-articular pain, such as nerve-related pain, were examined for the effects of interventions that included treatments like injection therapy. Studies were also excluded if the association between an exercise intervention and an outcome was not examined (for clinical trials, cohort studies, or case series) or if the paper focused on postoperative management.

### 2.4. CERT Assessment

The CERT checklist was used to extract data and assess the reporting completeness of the included studies [9]. The CERT comprises 16 items across seven domains: what (materials), who (provider), how (delivery), where (location), when/how much (dosage), tailoring (what, how), and how well (compliance/planned and actual). Each item was scored as 0 (not described), 1 (described), or NA (not applicable). Scores range from 0 to 19, with higher scores indicating more thorough descriptions. For studies with multiple exercise therapy groups, the protocol deemed superior was analyzed. Data from each study and related sources (e.g., appendices, Appendix A, and development descriptions) were independently extracted and evaluated by two researchers experienced in treating SIS-related pain with exercise therapy. The Explanation and Elaboration statement for the CERT guided the scoring [12]. Reasons for items marked ’not described’ were noted. The details and location of each item response were recorded for each study (Appendix A).

### 2.5. Risk of Bias Assessment

Risk of bias for clinical trials and case series was independently assessed by two reviewers (FG, AGF) using the modified Downs and Black checklist [13]. The checklist comprises 27 items with yes/no/unable to determine options. These items cover different aspects: ten items for reporting overall study quality, three for external validity, seven for study bias, six for confounding and selection bias, and one for study power. Disagreements between reviewers were resolved by consensus. The maximum score on the checklist is 28, with item 5 scoring 2 if ’yes’, 1 if ’partially’, and 0 if ’no’. The total score reflects the count of ’yes’ responses, with ’no’ and ’unable to determine’ scoring zero. Each study received a total quality score based on the information provided. Studies were classified as high quality/low risk of bias (≥20), moderate quality/risk of bias (17–19), or low quality/high risk of bias (≤16). Higher scores in the modified Downs and Black checklist scores indicate less risk of bias. Any disagreements on CERT or modified Downs and Black scores were resolved through a consensus meeting, with a third researcher (JAP) providing the deciding vote if necessary.

### 2.6. Data Analysis

The data were analyzed using the IBM SPSS Statistics software Statistical Package of Social Sciences (SPSS v.29 for Windows). Cohen’s kappa was utilized to assess the inter-rater agreement on the CERT score and modified Downs and Black checklist score. Cohen’s kappa result has been interpreted as follows: values ≤ 0 indicate no agreement, 0.01–0.20 indicate none to slight agreement, 0.21–0.40 indicate fair agreement, 0.41–0.60 indicate moderate agreement, 0.61–0.80 indicate substantial agreement, and 0.81–1.00 indicate almost perfect agreement [14]. Median and inter-quartile ranges (IQRs) were employed to describe the data. A single researcher synthesized the item responses from the included studies to provide an overview of the intervention contents.

## 3. Results

### 3.1. Study Identification and Selection

Figure 1 illustrates the progression of studies through the review process. Ultimately, five studies were incorporated into the CERT synthesis. The CERT scores and characteristics of the studies are described for the five studies that were ultimately included in the synthesis.

### 3.2. Study Characteristics

#### 3.2.1. Study Design

Among the five studies, four were RCT [15,16,17,18], and one was a case series study [19]. The sample sizes ranged from 10 to 88 participants. The studies used exercise therapy alone or in conjunction with manual therapy.

#### 3.2.2. Participants

The studies reported mean ages ranging from 17 to 43 with a diagnostic of SIS in overhead athletes [15,16,17,18,19].

#### 3.2.3. Outcome Measures and Results

The studies incorporated in the synthesis provided data on the effects of interventions measured through both shoulder-specific and general patient-reported outcomes, in addition to assessments of physical function (Table 1).

#### 3.2.4. CERT Score Synthesis

None of the studies mentioned CERT in their methodology section. CERT scores ranged between 6 and 13, with a median of 8 points (interquartile range: 1). Five studies [15,16,17,18,19] reported on at least three CERT items. The most frequently reported items were equipment (item 1) and tailoring (item 14), each reported in all five studies. The least reported items included motivation strategies (item 6), adherence (item 5), and fidelity (item 16), with none of the studies reporting on these items. Details of the scores can be found in Table 2, and the protocol content is available in Appendix A. The inter-rater reliability for the CERT scores showed a Cohen’s Kappa value = 0.85, indicating perfect agreement between the two raters.

#### 3.2.5. CERT Item Synthesis

WHAT (ITEM 1): All five studies [15,16,17,18,19] described the equipment used. Commonly used materials included resistance bands [15,16] and dumbbells. TRX or slings [17] were used in one study.

WHO (ITEM 2): Three studies [15,16,18] sufficiently described the title and qualifications of the prescriber (physical therapists), while two studies [17,19] provided no details about the physical therapists.

HOW (ITEM 3-11)

INDIVIDUAL/GROUP (ITEM 3): One study [19] provided details on whether exercise therapy was conducted in individual training sessions.

SUPERVISED/UNSUPERVISED (ITEM 4): Three studies [16,18,19] reported on supervision. One of these [19] used a non-supervised exercise session (home-based), while two studies [16,18] described a supervised exercise intervention by the physical therapist.

ADHERENCE (ITEM 5): None of the studies reported tracking adherence.

MOTIVATION (ITEM 6): None of the studies reported any motivation strategies used.

PROGRESSION (ITEMS 7A&B): Five studies [15,16,17,18,19] detailed the criteria for the progression of exercise with a variety of methods, reporting a rate of perceived exertion [17], a visual analog scale pain cut-off [19], or the use of time frames [15,16].

Three studies [15,16,17] described how the program was progressed. Most studies employed concurrent methods of progression, which included increasing exercise volume through additional repetitions and/or sets [15,16,17], enhancing intensity by targeting heavier loads (mostly with heavier elastic bands) [15,16], and advancing from isolated muscle exercises to more complex movements, such as compound functional movements [15,16].

EXERCISES (ITEM 8): Four studies [15,16,18,19] reported the exercises used. These included an isolated range of motion exercises (mobility) [15,16], shoulder and scapular exercises with weights or elastics bands in various positions [15,16,18,19], or upper extremity bodyweight exercises (such as quadruped push-up plus camel or chair press) [15,16].

HOME COMPONENT (ITEM 9): One study [19] included a home component in their exercise program, primarily offering participants a home-based regimen. Supervised sessions were conducted to monitor exercise technique and progression. This study used the home program as the sole intervention.

NON-EXERCISE COMPONENT (ITEM 10): Two studies [15,16] described non-exercise components, such as manual therapy (thoracic and/or shoulder mobilizations). No studies used patient education or activity modification.

ADVERSE EVENTS (ITEM 11): One study [19] reported adverse events related to their exercise intervention, specifically noting an increase in pain levels.

WHERE (ITEM 12): Two studies [18,19] included descriptions of the study setting: one implemented a home-based intervention [19], while the other was conducted in a university laboratory [18].

WHEN/HOW MUCH (ITEM 13): Five studies [15,16,17,18,19] reported on intervention dosage, with durations ranging from thirty minutes to eight weeks, and frequency ranged from daily training to three sessions per week. Five studies [15,16,17,18,19] provided dosage based on a measure of intensity, such as the rate of perceived exertion or a percentage of repetition maximum (RM).

TAILORING (ITEMS 14A&B): Five studies [15,16,17,18,19] included whether the program was tailored to the participants, with three using an individualized treatment [17,18,19] and two using a generic exercise program [15,16]. The treating physiotherapist adapted the program according to the patient’s impairment, pain-free range of motion, and sport-specific demands.

STARTING LEVEL (ITEM 15): Three studies [17,18,19] reported the patients’ starting level: two studies [17,19] utilized repetition maximum-based starting levels, and one adjusted the initial dose based on patient rating perceived exertion.

HOW WELL (ITEM 16A and 16B): None of the studies reported fidelity. One study [19] reported whether the interventions were implemented as planned, primarily using adherence reports and session attendance to detail the applied intervention.

### 3.3. Risk of Bias

The modified Downs and Black checklist scores for the five studies ranged from 13 (46%) to 24 (86%) out of a possible 28 (Table 3). One study was classified as having a low risk of bias [16], three studies [15,18,19] were classified as having a moderate risk of bias, and one study [17] was classified as having a high risk of bias. Most methodological shortcomings were related to external validity (items 11, 12, 13), internal validity study bias (items 14, 15), and confounding selection bias (items 21, 22, 23, 24, 25, 26). The included studies varied in design, with four RCTs [15,16,17,18] and one case series [19]. The agreement for risk of bias was almost perfect (Cohen’s Kappa = 0.81) (Appendix A).

## 4. Discussion

A total of five studies were included in the CERT synthesis that used therapeutic exercise as an intervention to treat SIS in overhead athletes. The maximum CERT score is 13 out of a possible 19. This indicates that studies using therapeutic exercise protocols for overhead athletes with SIS do not provide sufficient detail on their intervention, making it challenging to replicate and utilize these in clinical practice.

In accordance with the findings of our study, other investigations have employed the CERT checklist for the evaluation of interventions for patients with Achilles rupture [20], low back pain [21], hip osteoarthritis [22], and hip-related pain [23]. A systematic review utilizing the CERT checklist for patients with rotator cuff disorder reported a median CERT score of 5 (range 0–16) [24]. The items with the highest scores were equipment (item 1, with 100%) and tailoring (item 14, with 100%), while the lowest scores were adherence (item 5, with 0%), motivation (item 6, with 0%) and fidelity (item 16, with 0%). Motivation is one of the fundamental components of the patient’s belief that he can recover his function. Furthermore, motivation has been demonstrated to enhance adherence to treatment over the long term, thereby improving the patient’s prospects of recovery [25]. Adherence has a direct impact on the fidelity of the exercise program. Without consideration of adherence, the assessor or physiotherapist who prescribes the exercise program is unable to ascertain whether the prescribed exercises are performed with motivation, thereby potentially limiting the placebo effect of the exercises on the patient’s symptoms [26]. Additionally, 40% of the evaluated interventions did not consider the starting level (item 15 with 40%). Accounting for the patient’s baseline level before developing the therapeutic plan is a crucial aspect influencing its efficacy [27]. Our findings align with the systematic review conducted by Major et al. regarding the aforementioned items [24]. Considering the findings presented here and those of Major et al., it is evident that the lack of detail in the description of the interventions hinders their replication and utilization in clinical practice.

The optimal exercise type, dosage, duration, and expected outcomes remain unknown [6]. Furthermore, Tooth et al. explain in their systematic review that a glenohumeral internal rotation deficit of the joint, as well as an imbalance in the ratio between internal and external rotators, are risk factors for overuse shoulder injury in overhead athletes [28]. To modify these risk factors, it is necessary to produce new, standardized, and reproducible research protocols with the aim of reducing the incidence and prevalence of injury.

Despite the publication of the CERT checklist in 2016, Wright et al. did not employ it in their systematic review evaluating exercise prescriptions for overhead athletes. However, the authors of the study report that exercise-based interventions rely on expert opinion [10]. Furthermore, the analysis revealed that no article published after 2016 mentioned the use of the CERT checklist to program their exercise intervention. It can be concluded that the CERT checklist is still not being used to develop exercise programs for overhead athletes with SIS.

In the present study, the Modified Downs and Black checklist was employed for the assessment of methodological quality, rather than the RoB 2 tool, which is more appropriate for the assessment of RCTs. This decision was made due to the inclusion of a case series in the analysis, which comprised one of the five articles. The quality of the studies was found to be as follows: one was of good quality, three were of moderate quality, and one was of poor quality. The results reveal a dearth of rigorous design in studies examining the impact of SIS on overhead athletes. The limited number of studies included in this review, along with the methodological heterogeneity between RCTs and case series, poses a challenge in interpreting the CERT scores and assessing the overall risk of bias. The inclusion of a case series inherently lowers the level of evidence and may influence the comparability of results across studies. Moreover, the variations in study design impact the consistency in reporting exercise interventions, which may affect the replicability and generalizability of our findings. Future research should aim for higher methodological consistency by employing more robust study designs with larger sample sizes to enhance the reliability of CERT-based assessments.

The CERT checklist was developed with the objective of standardizing and improving the method by which exercise protocols are reported in clinical trials [9]. This facilitates more accurate interpretation, replication, and application of exercise programs [9]. Although the aim of this tool is to improve the quality of exercise interventions, certain shortcomings could be put forward. As we explained earlier, some items are more important than others. For instance, motivation, adherence, and starting level are factors that will determine the efficacy of the rehabilitation program for overhead athletes with SIS and should be accorded a higher score than the other items. Another crucial item mentioned previously is the description of the exercises, which could be divided into several sub-items, including, for example, dosage (in terms of volume or intensity), recoveries, and the way the exercises are carried out. Consequently, the revision and updating of the CERT checklist could be a means of enhancing the quality of exercise interventions.

One of the strengths of our review is that the two assessors are both physiotherapists with extensive experience in the use of therapeutic exercises to treat patients with SIS. Additionally, they possess a comprehensive understanding of the sports performed by overhead athletes. Furthermore, the level of agreement between the two evaluators was found to be substantial agreement, which provides a reliable indication of the quality of the interventions under study.

### Limitations

This study is subject to several limitations. Firstly, the participants in three of the five selected studies were engaged in throwing sports. However, this was not specified in the other two studies. This approach ensures poor homogeneity within the study population in accordance with the Bern classification [11]. Secondly, the number of articles included in the analysis is limited due to the small number of articles using exercise interventions for the treatment of this specific population with this condition. The present review, therefore, provides an overview of the quantity and quality of studies utilizing exercise therapy for the treatment of SIS in overhead athletes.

It is, thus, imperative to propose more standardized and reproducible protocols utilizing CERT as a reference for their programming, with the aim of enhancing the quality of SIS treatments in overhead athletes. Furthermore, this conclusion is in accordance with the conclusions of the Bern Consensus [11]. This is the rationale behind our decision to conduct RCTs that meet the requirements of the Bern consensus and utilize the CERT checklist as a guide for protocol development.

## 5. Conclusions

Only one study (one out of five) fulfilled the conditions required by the CERT to propose an exercise intervention for SIS in overhead athletes. The median CERT score was 8 (IQR 1), with no study achieving the maximum score. Future research should adopt a more standardized approach to defining key parameters such as exercise intensity, volume, progression, and recovery periods. Additionally, adherence and motivation strategies should be systematically incorporated into study protocols to enhance patient engagement and optimize therapeutic outcomes. Researchers should also ensure that exercise descriptions include precise dosage and delivery methods to facilitate replication and clinical implementation.

## Figures and Tables

**Figure 1 jcm-14-01657-f001:**
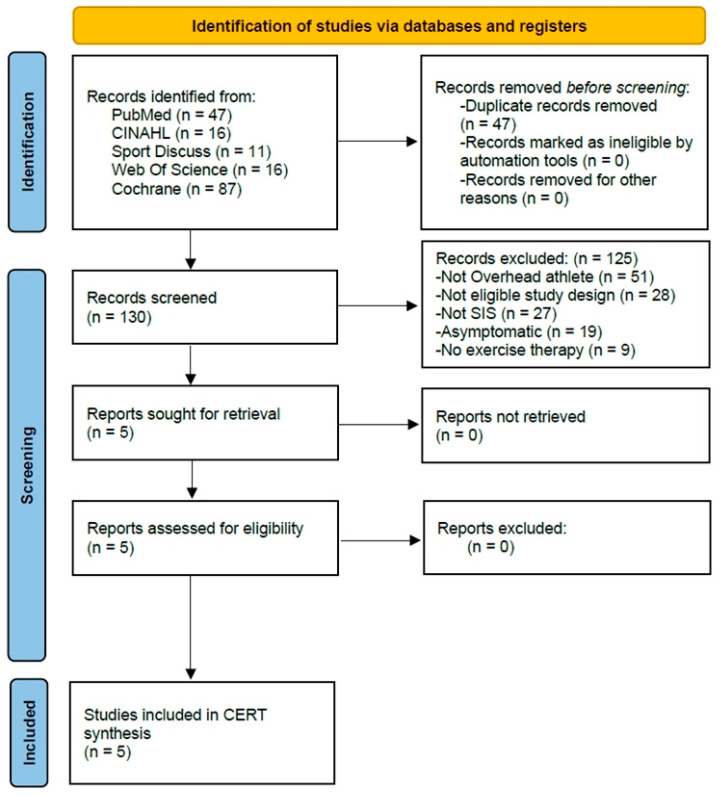
PRISMA flowchart illustrating the study selection process, including the number of articles screened, excluded, and ultimately included in the review.

**Table 1 jcm-14-01657-t001:** Characteristics of studies included in the CERT synthesis detailing study type, risk of bias, patient population, participants at baseline (PT/control), age (PT/control), interventions, outcomes, and results.

Included Studies	Study Type	Risk of Bias	Patient Population	Participants at Baseline, PT/Control (% Male)	Age, PT/Control (SD)	Intervention	Outcomes	Results
Sharma et al. 2021 [16]	RCT	Good	SIS in overhead athletes	44 (100)/44 (100)	21.30 (2.10)/21.80 (2.80)	resistance elastic band, stretching exercises, and mobilization of the thoracic and shoulder joints	isometric muscle strength (UT, MTr., LT, SA, A.D, Supr., and LD)	Increasing isometric strength of muscles for both groups and largest isometric strength improvement in PRE plus MT group.
Sharam et al. 2022 [15]	RCT	Fair	SIS in overhead athletes	5 (n.a.)/5 (NA.)	21.01 (1.04)/21.22 (1.56)	resistance elastic band, stretching exercises, and mobilization of the thoracic and shoulder joints	AHD measurement at 0°, 45°, 60°	AHD improvement was noted at all three levels only in the ET plus MT
De Mey et al. 2012 [19]	Case series	Fair	SIS in overhead athletes	40 (53)/NA	NA	Prone extension, forward flexion in side lying, external rotation in side lying, prone horizontal abduction with external rotation	SPADI and EMG of UT, MT, LT	SPADI scores significantly decreased, and LT showed earlier activation compared with UT and MT SA showed earlier activation compared with the UT, MT, and LT
Luo et al. 2024 [18]	RCT	Fair	SIS in overhead athletes	20 (90)/21 (77)	26.45 (4.13)/26.43 (5.55)	3 exercises to strengthen the LT and SA with EMG biofeedback	Corticospinal excitability was assessed using transcranial magnetic stimulation. Scapular kinematics and muscle activation during arm elevation were also measured.	Increase in motor-evoked potential in both group. Scapular-orientation group showed higher LT activation levels during arm elevation
Saadatian, 2022 [17]	RCT	Poor	SIS in overhead athletes	11 (NA) 11 (NA)/11 (NA)	28.18 (2.01)/27.09 (1.82)/29.09 (1.83)	OKS exercise in suspension with sling	JPS in ER, IR, and abduction of the dominant arm	change in the sling exercises group was higher in ER and IR, except abduction JPS than OKC group, than control groups

Abbreviatures: RCT, randomized controlled trial; SIS, subacromial impingement syndrome; PT, physical therapy; MT, manual therapy; AHD, acromiohumeral distance; UT, upper trapezius; MTr., middle trapezius; LT, lower trapezius; SA, serratus anterior; A.D, anterior deltoid; Supr., supraspinatus; LD, latissimus dorsi; SPADI, Shoulder Pain and Disability Index; EMG, electromyography; JPS, joint position sense; ER, external rotation; IR, internal rotation; OKS, open kinetic chain sling; OKC, open kinetic chain.

**Table 2 jcm-14-01657-t002:** CERT scores for included studies, presenting individual items and total scores across seven domains: what (materials), who (provider), how (delivery), where (location), when/how much (dosage), tailoring, and how well.

Author and Year	1. Equipment	2. Qualifications	3. Individual/Group	4. Supervision	5. Adherence	6. Motivation	7a. Progression Criteria	7b. Program Progression	8. Exercises	9. Home Component	10. Non-exercise	11. Adverse Events	12. Setting	13. Intervention	14a. Generic or Tailored	14b. How was it Tailored	15. Starting Level	16a. Fidelity	16b. Delivery as Planned	Total Sum
Sharma et al., 2022 [15]	1	0	0	0	0	0	1	1	0	0	1	0	0	1	0	0	0	1	0	**6**
Sharma et al., 2021 [16]	1	1	0	1	0	0	1	1	0	0	1	0	0	1	0	0	0	1	0	**8**
De Mey et al., 2012 [19]	1	0	1	1	0	0	1	1	1	1	0	1	1	1	1	1	1	1	1	15
Luo et al., 2024 [18]	1	1	0	1	0	0	0	0	1	0	0	0	1	1	1	1	0	1	0	9
Saadatian et al., 2022 [17]	1	0	0	0	0	0	1	1	0	0	0	0	0	1	1	1	1	1	0	**8**
Item Sum	5	2	1	3	0	0	4	4	2	1	2	1	2	5	3	3	2	5	1	

**Table 3 jcm-14-01657-t003:** Modified Downs and Black checklist results for risk of bias assessment, showing reporting, external validity, internal validity (study bias), internal validity for confounding (selection bias), and total scores for each study.

Scheme 28.	Reporting										External Validity			Internal Validity (Study Bias)							Internal Validity—Confounding (Selection Bias)						Power	Score (Out of 28)	Percent
	1	2	3	4	5	6	7	8	9	10	11	12	13	14	15	16	17	18	19	20	21	22	23	24	25	26	27		
Sharma et al., 2021 [16]	1	1	1	1	2	1	1	0	1	1	1	1	1	0	1	0	1	1	1	1	1	0	1	1	1	1	1	24	86
Sharma et al., 2022 [15]	1	1	1	1	2	1	1	0	1	1	0	0	0	0	1	0	1	1	0	1	1	0	1	0	1	1	1	19	68
De Mey et al., 2012 [19]	1	1	1	1	2	1	1	1	1	1	0	0	0	0	0	1	1	1	1	1	0	1	0	0	0	1	0	18	64
Luo et al., 2024 [18]	1	1	1	1	2	1	1	0	1	1	1	0	0	0	0	0	1	1	1	1	0	0	1	0	1	1	0	18	64
Saadatian et al., 2022 [17]	1	1	1	1	2	1	1	0	1	1	0	0	0	0	0	0	1	0	1	1	0	0	0	0	0	0	0	13	46

## Data Availability

The data that support the findings of this study are available upon request from the corresponding author, Jaime Almazán-Polo.

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
