# Peer review of "Therapeutic Exercise Prescription for Overhead Athletes with Shoulder Impingement Syndrome: A Systematic Review and CERT Analysis"

_jcm, 2025, doi:10.3390/jcm14051657_

Round 1

Reviewer 1 Report

Comments and Suggestions for Authors

This study is a systematic review that evaluates the reporting quality of exercise protocols for the treatment of shoulder impingement syndrome (SIS) in overhead athletes using the Consensus for Exercise Reporting Template (CERT) checklist. The study's objective is clear, and the approach of analyzing the impact of exercise prescription reporting quality on clinical reproducibility and applicability is meaningful. However, the following revisions are necessary:

  1. Introduction: The study mentions that previous research has relied on expert opinion, but it lacks a clear explanation of how this study improves upon past research.
  2. Introduction: While the study aims to evaluate exercise prescriptions using the CERT checklist, key CERT components (e.g., exercise intensity, frequency, and individualization) are not emphasized in the introduction, making the study’s focus unclear.
  3. Discussion: The study states that certain components (e.g., exercise intensity, frequency, and duration) were not adequately reported in previous studies, but it does not sufficiently discuss how these factors impact patient treatment outcomes.

Author Response

Reviewer 1:

This study is a systematic review that evaluates the reporting quality of exercise protocols for the treatment of shoulder impingement syndrome (SIS) in overhead athletes using the Consensus for Exercise Reporting Template (CERT) checklist. The study's objective is clear, and the approach of analyzing the impact of exercise prescription reporting quality on clinical reproducibility and applicability is meaningful. However, the following revisions are necessary:

Introduction: The study mentions that previous research has relied on expert opinion, but it lacks a clear explanation of how this study improves upon past research.

Response: Thank you for your comment. We have now expanded the Introduction section to explicitly highlight the gap in the literature and how our study provides a more structured evaluation of exercise interventions. Specifically, we emphasize that while prior research has relied on expert opinion, our study systematically assesses the quality of exercise prescriptions using the CERT checklist.

Introduction: While the study aims to evaluate exercise prescriptions using the CERT checklist, key CERT components (e.g., exercise intensity, frequency, and individualization) are not emphasized in the introduction, making the study’s focus unclear.

Response: Thank you for pointing out the need to emphasize key CERT components in the Introduction. We have now revised the section to explicitly mention exercise intensity, frequency, and individualization as fundamental aspects of the CERT checklist.

Discussion: The study states that certain components (e.g., exercise intensity, frequency, and duration) were not adequately reported in previous studies, but it does not sufficiently discuss how these factors impact patient treatment outcomes.

Response: Thank you for your consideration. We have included a paragraph according to your suggestion.

Reviewer 2 Report

Comments and Suggestions for Authors

The manuscript entitled “Therapeutic exercise prescription for overhead athletes with shoulder impingement syndrome: A Systematic Review and CERT Analysis” presented a systematic review evaluating the quality of exercise interventions for shoulder impingement syndrome in overhead athletes using the CERT checklist. The topic is clinically relevant, and the application of CERT to this population fills a notable gap in the literature. However, several areas require clarification and expansion to strengthen the manuscript.

  1. The authors should offer the search term of all databases. Please follow the PRISMA guidelines.
  2. The methods section should be rewrite.
  3. The results section should be organized more systematically.
  4. Define all abbreviations in tables (e.g., AHD, SPADI) at first mention or in footnotes to aid readability.
  5. Only five studies were included, one of which was a case series, which limits the generalizability of the study results. Although the authors have recognized this, they should explicitly discuss how the small sample size and the diversity of study designs (randomized controlled trials and case series) might affect the interpretation of the CERT score and the risk of bias in the discussion.
  6. The conclusion calls for standardized protocols but lacks actionable steps. Specific recommendations  should be provided.

Author Response

Reviewer 2:

The manuscript entitled “Therapeutic exercise prescription for overhead athletes with shoulder impingement syndrome: A Systematic Review and CERT Analysis” presented a systematic review evaluating the quality of exercise interventions for shoulder impingement syndrome in overhead athletes using the CERT checklist. The topic is clinically relevant, and the application of CERT to this population fills a notable gap in the literature. However, several areas require clarification and expansion to strengthen the manuscript.

  1. The authors should offer the search term of all databases. Please follow the PRISMA guidelines.

Response: Thank you for comment. The search strategy is in the appendix 2 (Supplementary material).

  1. The methods section should be rewrite.

Repsonse: Thank you for your suggestions. We considered your comment and have reviewed the methods section.

  1. The results section should be organized more systematically.

Response: Thank you for your suggestion. We have organized the results section.

  1. Define all abbreviations in tables (e.g., AHD, SPADI) at first mention or in footnotes to aid readability.

Response: Thank you for your comment. We have defined all abbreviations.

  1. Only five studies were included, one of which was a case series, which limits the generalizability of the study results. Although the authors have recognized this, they should explicitly discuss how the small sample size and the diversity of study designs (randomized controlled trials and case series) might affect the interpretation of the CERT score and the risk of bias in the discussion.

Response: We appreciate the reviewer’s comment regarding the impact of the small sample size and the diversity of study designs on the interpretation of the CERT score and the risk of bias. We have now explicitly discussed these limitations in the Discussion section, elaborating on how the inclusion of only five studies, one of which was a case series, may influence the robustness of our findings and the reliability of the CERT scores. We acknowledge that the heterogeneity in study designs (RCTs and case series) might introduce variability in the methodological quality and reporting standards, potentially affecting the generalizability of our results.

  1. The conclusion calls for standardized protocols but lacks actionable steps. Specific recommendations  should be provided.

Response: We acknowledge the reviewer's concern regarding the need for specific recommendations to guide the standardization of exercise protocols. In response, we have expanded the Conclusion section by outlining key actionable steps to improve the implementation of the CERT checklist in future research.

Round 2

Reviewer 1 Report

Comments and Suggestions for Authors

The authors revised all of them appropriately according to the review.

Reviewer 2 Report

Comments and Suggestions for Authors

The author has already addressed my concerns and I have no more questions.